# An End-to-End Graph Attention Network Hashing for Cross-Modal Retrieval

**Huilong Jin[1], Yingxue Zhang[2], Lei Shi[3,4*], Shuang Zhang[1*], Feifei Kou[5,6]**
**Jiapeng Yang[3], Chuangying Zhu[6], Jia Luo[7]**
[1] College of Engineering, Hebei Normal University
[2] College of Computer and Cyber Security, Hebei Normal University
[3] State Key Laboratory of Media Convergence and Communication, Communication University of China
[4] Key Laboratory of Education Informatization for Nationalities (Yunnan Normal University),
Ministry of Education
[5] School of Computer Science, Beijing University of Posts and Telecommunications
[6] Guangxi Key Laboratory of Trusted Software, Guilin University of Electronic Technology
[7] Chongqing Research Institute, Beijing University of Technology
*Correspondence: `leiky_shi@cuc.edu.cn`, `zshuang@hebtu.edu.cn`

## Abstract

Due to its low storage cost and fast search speed, cross-modal retrieval based on hashing has attracted widespread attention and is widely used in real-world applications of social media search. However, most existing hashing methods are often limited by uncomprehensive feature representations and semantic associations, which greatly restricts their performance and applicability in practical applications. To deal with this challenge, in this paper, we propose an end-to-end graph attention network hashing (EGATH) for cross-modal retrieval, which can not only capture direct semantic associations between images and texts but also match semantic content between different modalities. We adopt the contrastive language image pretraining (CLIP) combined with the Transformer to improve understanding and generalization ability in semantic consistency across different data modalities. The classifier based on graph attention network is applied to obtain predicted labels to enhance cross-modal feature representation. We construct hash codes using an optimization strategy and loss function to preserve the semantic information and compactness of the hash code. Comprehensive experiments on the NUS-WIDE, MIRFlickr25K, and MS-COCO benchmark datasets show that our EGATH significantly outperforms against several state-of-the-art methods.

## 1 Introduction

With the rapid development of information technology and social networks, a variety of modal data, including text, images, audio, and video, have gradually integrated into our daily lives. Due to the differences in the amount of information and expression of different modalities, how to retrieve information related to another modality in one modality has become a research hotspot in the field of information retrieval [1][2][3]. Traditional methods rely on manually designed feature extraction and similarity measurements, but it is difficult to adequately capture the complexity and diversity of data when dealing with large-scale high-dimensional data. Cross-modal hashing retrieval [4][5][6][7], due to its fast retrieval speed and high storage efficiency, has received widespread attention.

The existing cross-modal hashing methods are mainly divided into two types: supervised methods [8][9][10][11][12] and unsupervised methods [13][14][15][16]. Unsupervised methods learn models by exploring data structures and distributions without relying on labeled data. It has low

retrieval accuracy in complex cross-modal scenarios due to a lack of semantic alignment. In contrast, supervised methods leverage labeled data to establish the associations between different modalities and therefore accurately capture the similarities between different modalities, thus improving the accuracy of retrieval. Despite the wide application of supervised methods in cross-modal retrieval, there are some challenges that still remain. Traditional feature extraction techniques tend to focus on local features, which can hinder the capture of global information in images, ultimately affecting semantic consistency across modalities. Additionally, the high computational complexity associated with constructing label co-occurrence matrices limits their applicability to large-scale labeled datasets. It is also crucial for the hashing module to preserve the semantic information of the original data while maintaining compactness.

To address these challenges, we propose a novel supervised cross-modal hashing method based on graph attention network, termed end-to-end graph attention network hashing (EGATH) method. Our approach integrates the contrastive language image pretraining (CLIP) and transformer models to facilitate the global extraction of features from both images and texts. We implement a label classifier module utilizing graph attention networks (GAT) to enrich the semantic depth of feature representations while reducing the computational complexity associated with labeling. In addition, we constructed two functions to control the range of hash code values: a cosine similarity function to ensure the compactness and semantic preservation of hash codes, and a ternary loss function to bolster the robustness of hash codes. The main contributions of our work are as follows:

- By combining CLIP and transformer technologies, an end-to-end architecture is implemented that significantly improves the model's ability to capture global features of multi modal data, thus ensuring semantic consistency of images and text. Unlike other existing works, where CLIP and transformer are both used for text and image to extract features. We utilize CLIP coupled with transformer to extract features for image data, and feature extraction via a transformer for text, which can realize lightweight network.
- Predicted labels are combined with the feature extracted from the feature modules of image net and text net to enhance feature representation. We utilize GAT as a label classifier to explore the hidden information in the label to predict labels, which can directly model the label graph to dig the correlation between labels and has higher flexibility than other label classification using preset weights for feature processing.
- Our EGATH was evaluated on three widely recognized benchmark datasets: NUS-WIDE, MIR-Flickr25K, and MS-COCO, demonstrating clear performance advantages. Experimental results show that our method outperforms current state-of-the-art cross-modal hashing methods.

## 2  Methodology

### 2.1  Notations and Problem Definition

In this paper, we introduce the notation used as follows. Suppose we have a training dataset $\mathbf{O} = \{(x_i, y_i)|i \in [1, n]\}$, where $\mathbf{x}_i \in \mathbb{R}^{1 \times d_I}$ represents the data of the $i^{\text{th}}$ image sample and $\mathbf{y}_i \in \mathbb{R}^{1 \times d_T}$ represents the data of the $i^{\text{th}}$ text sample. Here, $n$, $d_I$, and $d_T$ represent the number of samples, the dimension of image features, and the dimension of text features, respectively. We use $\mathbf{F}_I \in \mathbb{R}^{d_I}$ and $\mathbf{F}_T \in \mathbb{R}^{d_T}$ to denote the extracted features of images and texts. Given a label set $\mathbf{L} = \{\mathbf{I}_i|i \in [1, N]\}$, where $\mathbf{L}$ is a set of labels composed of label $\mathbf{I}_i$, and $i$ ranges from 1 to $N$, $N$ represents the number of label categories. The label co-occurrence matrix is constructed based on the frequency of label co-occurrence in the dataset, and after thresholding, the adjacency matrix $\mathbf{A}$ is obtained. The labelled glove [17] vector $\mathbf{W}$ and adjacency matrix $\mathbf{A} \in \mathbb{R}^{d_l \times d_l}$ is constructed as input to GAT, where $d_l$ represents the dimension of the labeled feature. This paper aims to obtain compact hash codes $\mathbf{B}_I \in \{-1, 1\}^h$ and $\mathbf{B}_T \in \{-1, 1\}^h$, with $h$ representing the length of the hash codes. The $\tanh(\cdot)$ function and $\text{sign}(\cdot)$ function are used to constrain the values of the hash codes.

### 2.2  Network Architecture of EGATH

The EGATH framework depicted in Figure 1 encompasses two streams, designed for learning the hash functions for image and text modalities. For image data, we utilize CLIP coupled with a transformer to extract features, while for text data, we initially represent it as a bag-of-words (BoW) vector, followed by feature extraction via a transformer. To enhance the discriminative power of the representations for both types of data, we employ a classifier module based on GAT to integrate the

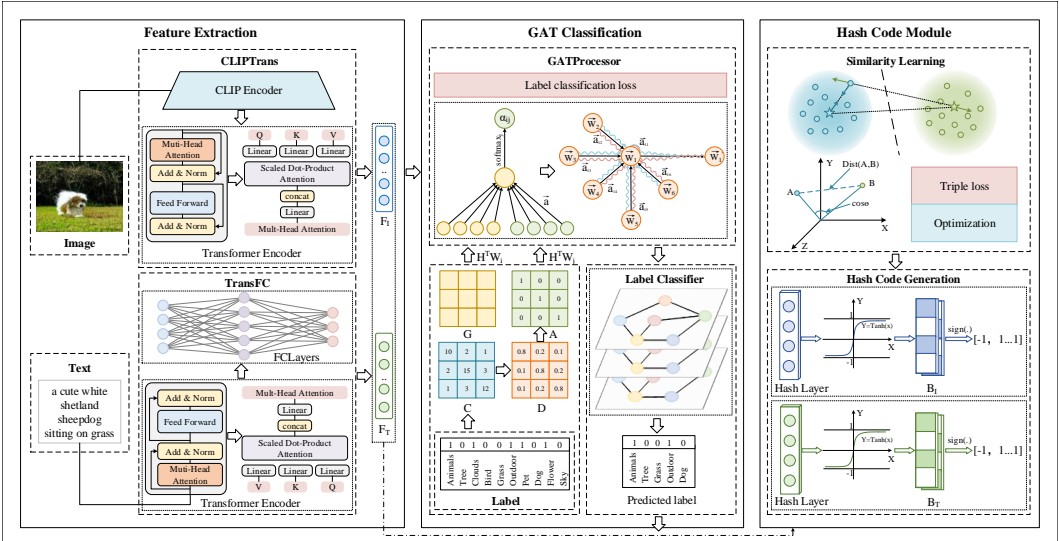

Figure 1: Our proposed EGATH framework consists of three main modules: 1) a feature extraction module, which employs the CLIP and transformer encoders to extract the global features of each modality; 2) a GAT classification module, which uses a graph attention network to dig deeper into the structural and semantic information of the labels; 3) a hash code module, which employs two functions and an optimization strategy to generate a hash code that preserves semantic information.

co-occurrence information of labels into the word embeddings. This process, facilitated by a dynamic attention mechanism, ensures that the learned word embeddings can reflect the interrelationships between labels, which are then used to predict the labels of instances in different modalities. In this manner, the predicted labels not only enrich the semantic representation of features but also enhance their distinctiveness. Thanks to the capabilities of the GAT classifier module, it effectively bridges the information gap when dealing with and integrating information from two distinct types of data, fostering deeper semantic consistency and enhanced representational power between different modalities. Finally, under the hash code loss function, we obtain hash codes that preserve more relevant semantic information, thereby improving the quality of retrieval.

## 2.3 Feature Extraction

In the EGATH model, the feature extraction part primarily consists of two components: an image network and a text network.

### 2.3.1 CLIPTrans

We adopt the CLIP[18] to map images and text into a shared vector space to understand the semantic associations between the two modalities. The input image is denoted as $\mathbf{x}_i$, and the specific feature extraction process can be expressed as follows:

$$f_I = \text{CLIP}(\mathbf{x}_i) \tag{1}$$

While the image features extracted by the CLIP model already contain rich information, their dimensions are not suitable for subsequent processing steps. To resolve this issue, we use a transformer encoder to perform dimension expansion and alignment on the features. The specific dimension expansion and alignment process is expressed as follows:

$$\text{Attention}(\mathbf{Q}, \mathbf{K}, \mathbf{V}) = \text{softmax}\left(\frac{\mathbf{Q}\mathbf{K}^{\mathbf{T}}}{\sqrt{d}}\right)\mathbf{V} \tag{2}$$

$$\mathbf{Q} = f_I \mathbf{W}^{\mathbf{Q}}, \quad \mathbf{K} = f_I \mathbf{W}^{\mathbf{K}}, \quad \mathbf{V} = f_I \mathbf{W}^{\mathbf{V}} \tag{3}$$

The terms $W^Q \in \mathbb{R}^{d \times d_I}$, $W^K \in \mathbb{R}^{d \times d_I}$, and $W^V \in \mathbb{R}^{d \times d_I}$ represent the weight matrices that, when multiplied by the feature $f_I$, yield the queries, keys, and values. The self-attention output provides a

feature representation enriched with contextual information.

$$\text{FFN}(x) = \max(0, x\mathbf{W}_1 + b_1)\mathbf{W}_2 + b_2 \tag{4}$$

$$\mathbf{F}_I = \text{FFN}(x)\mathbf{W}_{\text{output}} + b_{\text{output}} \tag{5}$$

The output of the self-attention layer is processed by a feedforward network, where $\mathbf{W}_1$, $\mathbf{W}_2$, and $\mathbf{W}_{\text{output}}$ are the weight matrices for the linear and output layers of the feedforward network, respectively. $b_1$, $b_2$, and $b_{\text{output}}$ are the corresponding bias terms, and $\mathbf{F}_I$ is the dimension of the image features after adjustment and alignment.

### 2.3.2 TransFC

In our text network, we utilize the transformer model to extract textual features, resulting in a high-dimensional feature representation denoted by $f_T$. Given the high dimensionality of the extracted features, a fully connected layer is introduced to adjust the feature dimensions appropriately, thereby enhancing the model's capability to capture complex features and patterns.

### 2.4 GAT Classifier

In this section, we will delve into the label classifier module based on GAT. GAT incorporates co-occurrence information of labels into word embeddings and uses the learned embeddings to predict labels across modalities. The predicted labels are further combined with feature representations to enhance their discriminative properties.

We construct an adjacency matrix based on the co-occurrence frequency of labels. Firstly, we analyze the number of times label pairs appear in the training set to obtain the matrix $\mathbf{C}$, where $\mathbf{C}_{ij}$ denotes the number of occurrences of label $i$ and label $j$ together. Initially, we sum across the rows or columns of the co-occurrence matrix, and the marginal probability of label $i$ can be represented as:

$$\mathbf{N}_i = \sum_{j=1}^{N} \mathbf{C}_{ij} \tag{6}$$

where $\mathbf{N}_i$ is the total occurrence of label $i$ , including its co-occurrence with all other labels, then the conditional probability matrix $\mathbf{D}$ is calculated, where $\mathbf{D}_{ij}$ represents the probability of label j occurring given label i has occurred. The specific conditional probability can be expressed as follows:

$$\mathbf{D}_{ij} = \frac{\mathbf{C}_{ij}}{\mathbf{N}_i} \tag{7}$$

We set a threshold $\zeta$ to eliminate those marginal connections that may be caused only by noise. The main operation is as follows:

$$\mathbf{A}_{ij} = \begin{cases} 0, & \text{if } D_{ij} < \zeta \\ 1, & \text{otherwise} \end{cases} \tag{8}$$

According to $\mathbf{A}_{ij}$ in it is known that the set of neighbouring points j of data point i is $\mathcal{N}(i) = \{j \mid \mathbf{A}_{ij} \neq 0, 1 \leq j \leq n\}$. Inspired by the work [19], For each label pair (i, j) calculate the attention coefficient is expressed as follows:

$$\alpha_{ij} = \frac{\exp\left(\text{LeakyReLU}\left(\mathbf{a}^\top\left[\mathbf{W}\mathbf{g}_i \parallel \mathbf{W}\mathbf{g}_j\right]\right)\right)}{\sum_{k\in\mathcal{N}(i)}\exp\left(\text{LeakyReLU}\left(\mathbf{a}^\top\left[\mathbf{W}\mathbf{g}_i \parallel \mathbf{W}\mathbf{g}_k\right]\right)\right)} \tag{9}$$

where $\mathbf{W}$ is a learnable weight matrix, $\mathbf{a}$ is the attention vector that is used to compute the correlation between pairs of labels, $\mathbf{g}_i$ and $\mathbf{g}_j$ denotes the glove vector of label i and j.

Using the computed attention coefficients, the feature vectors of the labels are weighted and summed to update the features of the labels, which are represented as follows:

$$\mathbf{H}_n = \sum_{j\in\mathcal{N}(i)} \alpha_{ij}\mathbf{W}\mathbf{g}_j \tag{10}$$

It is known from GAT that the output of the feature representation of the last layer is $\mathbf{H}_n, \hat{\mathbf{S}}_i$ where the predicted scores of the labels are represented as follows:

$$\hat{\mathbf{S}}_i = \mathbf{H}_n\mathbf{f}'_* \tag{11}$$

where * represents I or T, $\mathbf{f}'_*$ represents the representation of a single image or text, and the multi-label classification loss can be expressed as follows:

$$\mathcal{L}_{loss} = \sum_{i=1}^{N} \left( \mathbf{S}_i \log(\sigma(\hat{\mathbf{S}}_i)) + (1 - \mathbf{S}_i) \log(1 - \sigma(\hat{\mathbf{S}}_i)) \right) \tag{12}$$

where $N$ represents the number of labels in the sample, summing over all $N$ labels to calculate the total loss of the sample, $\mathbf{S}$ represents the true value of the sample labels, $log(\sigma(\hat{\mathbf{S}})$ is the predicted score after applying the sigmoid function, mapping the original score to the (0,1) interval, log is used to compute the cross-entropy, sending the predicted label results to the hash layer for the next operation.

## 2.5 Hash Code Module

In this section, we delve deeper into the hash code module in the model. The primary function of this module is to map continuous features to discrete hash codes. This module achieves effective encoding of multimodal features and labels by combining the tanh(·) and sign(·) functions, as well as a strategy that combines triplet loss with cosine similarity.

### 2.5.1 Hash Function

During the training process, the phenomena of gradient vanishing and unbalanced distribution of hash codes can occur. We introduce a tanh(·) function after the hash coding layer, which effectively compress the input values into the range of -1 to 1. Meanwhile, tanh(·) provides relatively smooth gradient information, which is conducive to the updating and optimization of network parameters during the training process, defined as follows:

$$\mathbf{B}_I = \tanh(\mathbf{W}_I \cdot I + \mathbf{b}_I) \tag{13}$$

$$\mathbf{B}_T = \tanh(\mathbf{W}_T \cdot T + \mathbf{b}_T) \tag{14}$$

where $\mathbf{W}_I$ and $\mathbf{W}_T$ are the weights for mapping image and text features to the hash space, and $\mathbf{b}_I$ and $\mathbf{b}_T$ are the bias terms. Although the output of the hash layer is compressed to a certain range, it remains continuous binary values. In order to obtain discrete binary hash codes for retrieval, the sign(·) threshold function is applied to convert the continuous values into binary values, generating the final required hash code. The sign(·) function is defined as sign(x), and the following is the mathematical representation involved in the hash code module:

$$\text{sign}(x) = \begin{cases} 1, & \text{if } x \geq 0 \\ -1, & \text{if } x < 0 \end{cases} \tag{15}$$

where $x$ represents the values of $\mathbf{B}_I$ and $\mathbf{B}_T$. The processed hash codes are distinguished into positive and negative values, establishing a consistent hash space across different modalities of data, thereby obtaining discrete binary hash codes.

### 2.5.2 Loss Function

We use a combination of triplet loss [20] and cosine similarity to optimize the encoding process of the hash code, enabling the model to generate more accurate hash codes. Specifically, the cosine similarity is used to calculate the similarity between two hash vectors, and then the triplet loss function is used to guide the model training so that the cosine similarity of the positive sample pair is as close to 1 as possible, while the cosine similarity of the negative sample pair is less than a preset threshold, with the cosine similarity defined as follows:

$$\text{sim}(\mathbf{a}, \mathbf{b}) = \frac{\langle \mathbf{a}, \mathbf{b} \rangle}{\|\mathbf{a}\| \cdot \|\mathbf{b}\|} \tag{16}$$

where, $\mathbf{a}$ and $\mathbf{b}$ represent the hash code vectors of images and texts, respectively, with '·' denoting the dot product, and $\|\mathbf{a}\|, \|\mathbf{b}\|$ representing the norm of the hash code vectors. The triplet loss function is designed with three sample points: an anchor (Anchor) $\mathbf{a}$, a positive sample (Positive) $\mathbf{p}$, and a negative sample (Negative) $\mathbf{n}$. The anchor and the positive sample are highly similar, while the negative sample is dissimilar to the anchor. The main purpose of the triplet loss is to bring the anchor

and positive sample closer together while pushing the anchor and negative sample further apart, thereby preserving similarity in the representation space. The specific loss function is expressed as follows:

$$\mathbf{L}_1 = \|\mathbf{f}(\mathbf{a}) - \mathbf{f}(\mathbf{p})\|^2 \tag{17}$$

$$\mathbf{L}_2 = \|\mathbf{f}(\mathbf{a}) - \mathbf{f}(\mathbf{n})\|^2 \tag{18}$$

$$\mathcal{L}_{tri} = \max(\mathbf{L}_1 - \mathbf{L}_2 + \alpha, 0) \tag{19}$$

Among them $\mathbf{L}_1$ represents the squared distance calculated between the anchor $\mathbf{a}$ and the positive sample $\mathbf{p}$, while $\mathbf{L}_2$ represents the squared distance calculated between the anchor $\mathbf{a}$ and the negative sample $\mathbf{n}$. The parameter $\alpha$ is used to specify the minimum distance difference between the positive and negative samples.

## 2.6 Optimization

In this section, we discuss the optimization problem, mainly focusing on the optimization of hash codes and the program. We adopt the approach of GCDH [21] for iterative optimization of hash codes, introducing auxiliary variables $\mathbf{R}$, $\mathbf{Q}$, $\mathbf{P}$, and $\mathbf{B}$. Here, $\mathbf{R}$ represents the mapping from the feature space to the hash code space, mapping high-dimensional feature vectors to low-dimensional hash codes. $\mathbf{P}$ represents the target hash code with constraints, $\mathbf{Q}$ is used to ensure consistency between the target hash code and the current hash code, balancing the relationship between the two, and $\mathbf{B}$ represents the current hash code.

Hash learning is typically a non-convex optimization problem that cannot be solved for a global optimum in a single step. Therefore, it requires an iterative approach to gradually approximate the optimal solution. Initially, based on the current hash encoding $\mathbf{B}$, optimization from features to hash code mapping is performed to find a more suitable hash function. The update strategy is as follows:

$$\mathbf{R} = (\mathbf{L}^\top \mathbf{L} + \lambda \mathbf{I})^{-1} \mathbf{L}^\top \mathbf{B} \tag{20}$$

The update of $\mathbf{R}$ may cause a deviation between the hash representation of the data and the label, therefore, it is necessary to update $\mathbf{B}$ again to correct this deviation:

$$\mathbf{B} = \text{sign}(\mathbf{L}\mathbf{R} + \gamma(\mathbf{P} - \mathbf{Q}/\gamma) + \mu(\mathbf{B_I} + \mathbf{B_T})) \tag{21}$$

$\mathbf{P}$ is typically updated to better meet the constraints that $\mathbf{B}$ should adhere to. It can be viewed as an ideal state, and the hash code $\mathbf{B}$ should be as close as possible to this state. $\mathbf{B}$ is usually derived from the singular value decomposition of the hash code, so first, $\mathbf{B}$ is subjected to singular value decomposition, expressed as follows:

$$\mathbf{B} = \mathbf{U}\Sigma\mathbf{V}^T \tag{22}$$

where $\mathbf{U}$ and $\mathbf{V}$ are orthogonal matrices, containing the left and right singular vectors, respectively. We use the vector data from these matrices to update $\mathbf{P}$, with the main construction expressed as follows:

$$\mathbf{P} = \mathbf{U_p}\Sigma_p\mathbf{V_p^T} \tag{23}$$

where $\mathbf{U}_p$ and $\mathbf{V}_p$ are the columns selected from $\mathbf{U}$ and $\mathbf{V}$ that correspond to the largest singular values, and $\mathbf{V}_p^T$ is the diagonal matrix that includes the selected largest singular values. $\mathbf{Q}$ is introduced to gradually correct the deviation between $\mathbf{B}$ and $\mathbf{P}$. When $\mathbf{B}$ equals $\mathbf{P}$, the update of $\mathbf{Q}$ is shown as follows:

$$\mathbf{Q} = \mathbf{Q} + \gamma(\mathbf{B} - \mathbf{P}) \tag{24}$$

Throughout the optimization process, the updates of $\mathbf{P}$ and $\mathbf{Q}$ are carried out alternately. First, $\mathbf{P}$ is updated to reflect the constraints on the hash code $\mathbf{B}$, then $\mathbf{Q}$ is adjusted to correct the deviation from the target. This alternating update strategy enables the model to find the optimal hash code while satisfying constraints.

# 3 Experiments

In this section, we evaluate the performance of the proposed EGATH framework on three benchmark datasets using state-of-the-art methods. Then, we present a detailed discussion on ablation studies to investigate each component of our model.

Table 1: Experimental results on the three datasets.

| Task | Method | MIRFlickr25K | | | NUS-WIDE | | | MS-COCO | | |
|---|---|---|---|---|---|---|---|---|---|---|
| | | 16bit | 32bit | 64bit | 16bit | 32bit | 64bit | 16bit | 32bit | 64bit |
| I → T | DJSRH | 0.6652 | 0.6873 | 0.6987 | 0.5271 | 0.5582 | 0.6015 | 0.5257 | 0.5454 | 0.5646 |
| | JDSH | 0.7276 | 0.7426 | 0.7468 | 0.6536 | 0.6601 | 0.6900 | 0.5928 | 0.6348 | 0.6517 |
| | DCHMT | 0.8177 | 0.8221 | 0.8261 | 0.6711 | 0.6812 | 0.6932 | 0.6450 | 0.6331 | 0.6647 |
| | MLCAH | 0.7960 | 0.8080 | 0.8150 | 0.6440 | 0.6410 | 0.6430 | 0.5700 | 0.5620 | 0.5620 |
| | CDTH | 0.7317 | 0.7461 | 0.7477 | 0.6596 | 0.6613 | 0.6700 | 0.5853 | 0.6411 | 0.6573 |
| | UCCH | 0.7606 | 0.7620 | 0.7674 | 0.6718 | 0.6738 | 0.6891 | 0.6039 | 0.6249 | 0.6398 |
| | SCAHN | 0.8123 | 0.8131 | 0.8336 | 0.6588 | 0.6621 | 0.6669 | 0.6727 | 0.7108 | 0.7528 |
| | DSPH | 0.7935 | 0.8141 | 0.8363 | 0.6851 | 0.6996 | 0.7092 | 0.6466 | 0.6666 | 0.6757 |
| | **EGATH** | **0.8411** | **0.8562** | **0.8637** | **0.7191** | **0.7345** | **0.7480** | **0.7259** | **0.7688** | **0.7945** |
| T → I | DJSRH | 0.6710 | 0.6958 | 0.7043 | 0.5575 | 0.5680 | 0.5952 | 0.5590 | 0.5591 | 0.5519 |
| | JDSH | 0.7304 | 0.7326 | 0.7481 | 0.6439 | 0.6640 | 0.6921 | 0.5888 | 0.6510 | 0.6635 |
| | DCHMT | 0.8007 | 0.8021 | 0.8065 | 0.6852 | 0.6963 | 0.7009 | 0.6298 | 0.6176 | 0.6616 |
| | MLCAH | 0.7940 | 0.8050 | 0.8050 | 0.6620 | 0.6730 | 0.6870 | 0.5440 | 0.5470 | 0.5940 |
| | CDTH | 0.7315 | 0.7464 | 0.7503 | 0.6788 | 0.6815 | 0.6910 | 0.5846 | 0.6427 | 0.6573 |
| | UCCH | 0.7343 | 0.7342 | 0.7410 | 0.6740 | 0.6812 | 0.6945 | 0.6023 | 0.6258 | 0.6371 |
| | SCAHN | 0.7890 | 0.7971 | 0.8186 | 0.6718 | 0.6803 | 0.6980 | 0.7183 | 0.7504 | **0.8093** |
| | DSPH | 0.7928 | 0.8038 | 0.8149 | 0.6957 | 0.7147 | 0.7256 | 0.6473 | 0.6656 | 0.6774 |
| | **EGATH** | **0.8064** | **0.8185** | **0.8293** | **0.7270** | **0.7437** | **0.7539** | **0.7247** | **0.7729** | 0.8015 |

## 3.1 Datasets

To evaluate the proposed methods, we conducted experiments on three widely used cross-modal datasets, namely, MIRFlickr25K [22], NUS-WIDE [23], and MS-COCO [24]. The specific division of these datasets is outlined in Appendix B.

## 3.2 Implementation Details

In this paper, we use CLIP and transformer as feature extractors for images, transformer and Fully Connected Layer as feature extractors for text, and GAT as the label classifier to bridge the information gap between modalities. The input for GAT is the 300-dimensional glove vectors pre-trained on the Wikipedia dataset and the adjacency matrix between labels. We carefully set some hyperparameters $\alpha, \beta$, and k for auxiliary learning. Through experimental analysis, we examined the sensitivity of these parameters. Finally, we set the batch size to 64 and employed the Adam [25] optimization strategy for the main optimization, with a weight decay set to 0.0005 and adopting a method of dynamically adjusting the learning rate.

Our EGATH method is implemented in Pytorch [26], and the experiments were conducted on a server equipped with an NVIDIA GeForce RTX 3080 with 40GB of RAM.

## 3.3 Baselines and Evaluation Metrics

There are two cross modal hash retrieval tasks in this article (from text to image, from image to text), and we compared our method with eight advanced cross modal hash methods: mainly about DCHMT [27], JDSH [28], MLCAH [29], UCCH [30], DSPH [31], CDTH [32], SCAHN [33] and DJSRH [34]. To ensure fairness, we reran the code for the comparative experiments. For papers that did not provide code, we directly used the results reported in the papers, ensuring consistent dataset splits. In this study, we used three evaluation metrics to measure the performance of retrieval: mean average precision (mAP), the precision-recall curve (PR curve) , and top-K Precision curve (top-K curve). Detailed descriptions of these metrics are shown in Appendix C.

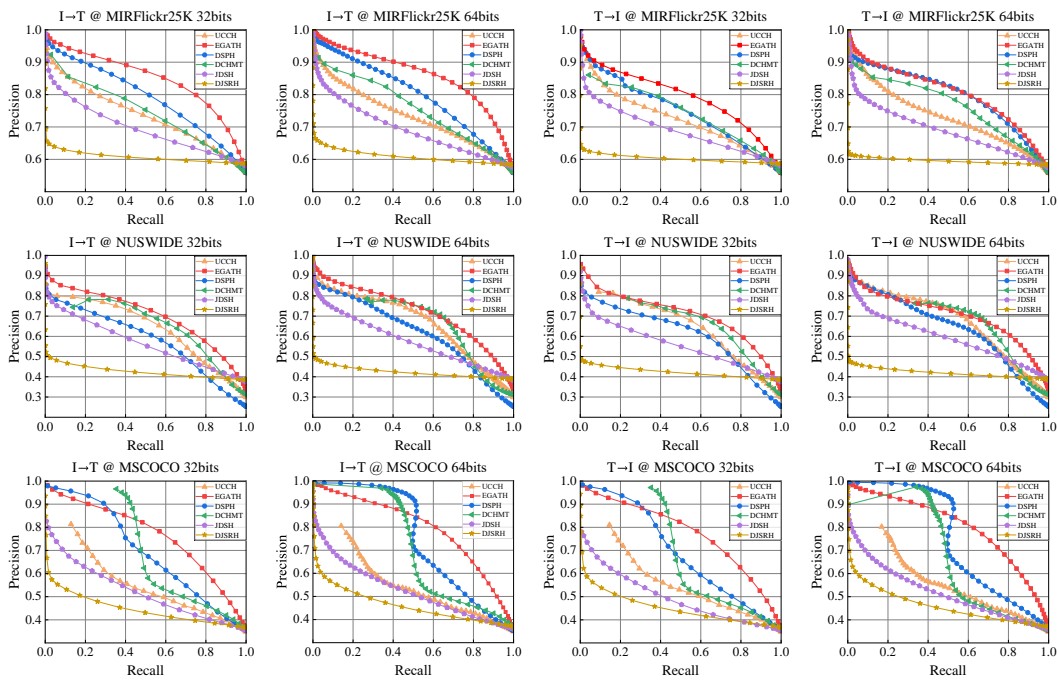

Figure 2: PR curves on the three datasets

## 3.4 Performance Comparison

To verify the effectiveness of the proposed EGATH method, we conducted performance comparisons on three datasets. All results are shown in Table 1, where "I → T" indicates image-to-text retrieval, and "T → I" represents text-to-image retrieval.

The optimal results from the experiment are shown in bold. Compared to the results of DSPH and DCHMT, especially the effect on MSCOCO dataset has a significant enhancement, which is due to the following reasons. The MSCOCO dataset with rich label information provides more contextual information for GAT, which effectively improves the accuracy of label prediction and the consistency of cross-modal features, and indicates that our method has a special advantage in dealing with data-rich datasets with diverse labels. For the coco dataset in SCAHN 64bit is higher than our results, which is due to the fact that SCAHN uses cross-media and intra-modal contrast against hashing mechanisms to enhance the differentiation of different modal semantic representations. Therefore, it achieves better performance in multi-label type datasets.

We introduce the PR curve in Figure 2. It can be observed that our model achieves high precision scores at all recall levels. This outperformance is due to the fact that EGATH significantly improves semantic discriminability between modalities and cross-modal consistency through feature networks, which enables the model more accurately identify and match relevant items in cross-modal data. In addition, GAT is used in EGATH to deeply mine the latent semantic information in the labels, which further enhances the model's performance in the feature representation and matching process.

Figure 3 demonstrates the performance of EGATH with different Top-K retrieval results. The K values include 50, 100, 200, 500, 1000, 1500, 2000, 2500, 3000, 3500, 4000, 4500, and 5000, and these experimental data mainly cover small-to large-scale retrieval tasks to evaluate the effectiveness of different methods under various retrieval scales. EGATH is consistently maintains a high accuracy rate under any top-K ordering. The trend in the figure shows that EGATH tends to be stable under different data sizes and K values. By combining aspects such as semantic matching and feature representation between different modalities, thus performing better in large-scale retrieval.

## 3.5 Ablation Study

To validate the effectiveness of our contributions, we designed two ablation experiments as follows:

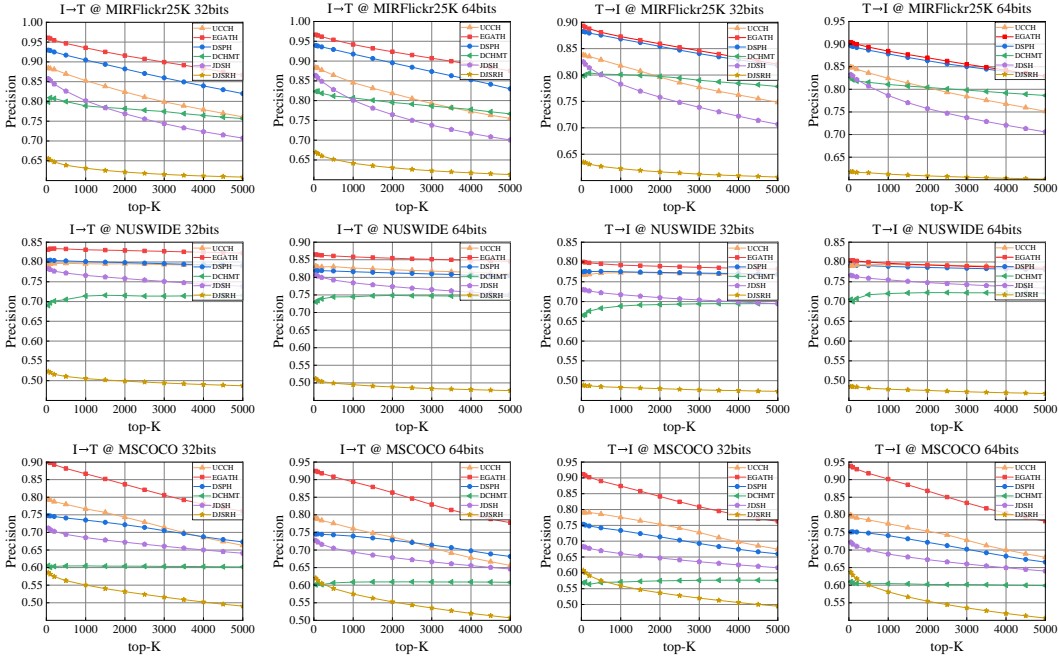

Figure 3: Top-K curves on the three datasets

- EGATH-G: This experiment removes the GAT classifier module, with all other modules being the same as in EGATH.
- EGATH-T: This experiment removes the triplet loss function based on cosine similarity, with all other modules being the same as in EGATH.

The main comparative results are presented in Table 2. The performance of the EGATH-G model is notably poor when handling imbalanced and sparse label data, underscoring the importance of the graph attention module. In the EGATH-T model, despite the absence of the hash code optimization strategy, the model retains some performance. However, compared to the complete EGATH model, its performance declines significantly, particularly under complex and diverse query conditions. The inclusion of the graph attention module effectively addresses label data challenges and enhances overall retrieval performance. Furthermore, the hash code optimization strategy contributes to generating more robust hash codes, which is crucial for improving retrieval accuracy and efficiency. The results from the ablation experiments clearly demonstrate that each component positively influences the overall performance.

In summary, these ablation experiments not only validate the effectiveness of the graph attention module and hash code optimization strategy but also showcase their significant roles in improving the performance of cross-modal retrieval systems.

Table 2: Ablation study on the three datasets.

| Task | Method | MIRFlickr25K | | | NUS-WIDE | | | MS-COCO | | |
|------|--------|--------|--------|--------|--------|--------|--------|--------|--------|--------|
| | | 16bit | 32bit | 64bit | 16bit | 32bit | 64bit | 16bit | 32bit | 64bit |
| I → T | EGATH-G | 0.8145 | 0.8354 | 0.8464 | 0.7072 | 0.7294 | 0.7429 | 0.6937 | 0.7333 | 0.7583 |
| | EGATH-T | 0.8348 | 0.8521 | 0.8622 | 0.7135 | 0.7297 | 0.7435 | 0.7029 | 0.7438 | 0.7736 |
| | **EGATH** | **0.8411** | **0.8562** | **0.8637** | **0.7191** | **0.7345** | **0.7480** | **0.7259** | **0.7688** | **0.7945** |
| T → I | EGATH-G | 0.7965 | 0.8102 | 0.8184 | 0.7163 | 0.7386 | 0.7477 | 0.6991 | 0.7372 | 0.7649 |
| | EGATH-T | 0.7990 | 0.8123 | 0.8224 | 0.7165 | 0.7390 | 0.7483 | 0.7058 | 0.7494 | 0.7785 |
| | **EGATH** | **0.8064** | **0.8185** | **0.8293** | **0.7270** | **0.7437** | **0.7539** | **0.7247** | **0.7729** | **0.8015** |

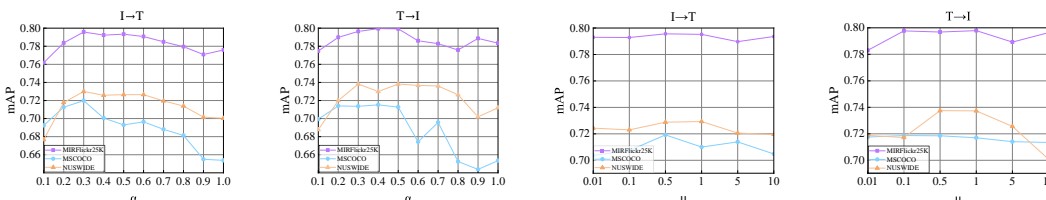

Figure 4: The influence of hyper-parameters

## 3.6 Parameter Sensitivity

In this study, we set several parameters for experimental optimization: $\alpha$=0.3, $\mu$=0.5. To analyze these parameters, we chose to validate on three datasets using a uniform 64-bit hash code. When adjusting the values of different parameters, the results of the mAP also fluctuate, confirming the rationality of our parameters for the model. The specific results are shown in Figure 4.

From this figure, we can see that the performance of cross-modal search improves with the increase of $\alpha$. When $\alpha$ is equal to 0.3, the optimal results are achieved for the mAP values on all three datasets, so we set the value of $\alpha$ to 0.3. When $\mu$ increases from 0.01 to 0.5, the performance metrics gradually improve, indicating that $\mu$ is helpful for the regularisation effect. However, when $\mu$ is increased to 5, the performance starts to deteriorate, probably because the regularisation is too strong, which limits the learning ability of the model, so we set the value of $\mu$ to 0.5.

## 4   Conclusion

This paper proposes an EGATH, which uses the CLIP model and transformer architecture to extract features from multimodal data, enhancing the semantic consistency between different modalities in an end-to-end manner. Furthermore, by integrating GAT, we strengthen the model's understanding of the data structure of different modalities, treating them as a set of interdependent object classifiers to deeply mine and learn the intrinsic graph structure features of the data. Labels are processed as word embeddings, and the predicted labels are obtained from these classifiers, enhancing cross-modal feature representation. We delve into the hidden features and relationships within the graph structure, achieving effective encoding and representation of data, and comprehensive experiments demonstrate the effectiveness of this method. Future work could explore the extension of EGATH to other types of multimodal data, such as audio and text, and investigate its scalability with larger datasets. Notably, this method is limited by the high computational complexity of the graph attention network, which constrains the model's scalability under large-scale label sets. Although cross-modal hashing retrieval enhances the convenience of information access and the potential for cross-cultural exchange, it also brings risks of privacy breaches and bias amplification. Therefore, strengthening data security and fairness management is essential to ensure its positive societal impact.

## Acknowledgments

This work is supported by the Science and Technology Project of Hebei Education Department (No. CXY2024050), Fundamental Research Funds for the Central Universities (No.2023RC08), the Guangxi Key Laboratory of Trusted Software (No.KX202304), National Natural Science Foundation of China (No.U22B2038, No.72104016), the 8th Young Elite Scientists Sponsorship Program by CAST (No.2022QNRC001), Open Research Subject of State Key Laboratory of Intelligent Game (No.ZBKF-24-12), the Natural Science Foundation of Chongqing, China (No.CSTB2023NSCQ-MSX0391), the Foundation of Key Laboratory of Education Informatization for Nationalities (Yunnan Normal University), Ministry of Education (No.EIN2024C006), and the Beijing Natural Science Foundation (No.9242003).

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

# Appendix / supplemental material

## A  Related Work

In this section, we introduce existing work related to our research, focusing primarily on the two forms of cross-modal hash retrieval.

### A.1  Supervised Cross-Modal Hashing

Supervised hashing methods [8][9][10][11][12][35] mainly utilize the semantic associations between different modalities and manually annotated label information to learn a function that can be mapped to a common hash space, enabling effective similarity search of data from different modalities in this hash space. Among these, semantic correlation maximization (SCM) [9] is a supervised multimodal hash method that uses all supervision information for training. It generates more distinctive hash codes by learning hash functions bit by bit and achieves high performance. Semantic preserving hashing (SePH) [8] is a supervised cross-view hashing method that mainly uses the semantic similarity of training data as supervision information and converts it into a probability distribution. Then, it approximates the semantic similarity between training data and the hash codes to be learned by minimizing the KL divergence. Batch learning of asymmetric discrete cross-modal hashing (BATCH) [36] uses collective matrix factorization to learn the common latent space of labels and different modalities and establishes a connection between the common latent space and hash codes with an asymmetric strategy by minimizing the distance-distance difference problem.

Although the above methods have achieved good results, with the integration of deep neural networks, many supervised deep cross-modal hashing methods have also achieved good performance[37]. Deep cross-modal hashing (DCMH) [38] proposes an end-to-end learning framework that integrates feature learning and hash code learning into one framework, enabling direct learning of discrete hash codes without relaxation. Self-supervised adversarial hashing (SSAH) [39] is an early attempt to incorporate adversarial learning into cross-modal hashing. It uses two adversarial networks to maximize the relevance between different modalities and employs a self-supervised semantic network to discover high-level semantic information.

### A.2  Unsupervised Cross-Modal Hashing

Regarding unsupervised cross-modal hashing, this method does not rely on manually annotated data. Instead, it learns the mapping relationships between different modalities by analyzing the intrinsic structure of the data, mapping data from different modalities to a common Hamming space. The goal is to bring similar samples closer together in the space, thereby improving retrieval performance. Collective matrix factorization hashing (CMFH) [40] uses collective matrix decomposition on data from different modalities of the same instance and adopts a latent factor model to learn unified hash codes. Cross-view hashing (CVH) [41] generates common hash codes based on the intra-modal and inter-modal similarities of multimodal data, effectively solving cross-view search problems.

Beyond the aforementioned shallow methods, the introduction of deep learning technologies has brought new possibilities for retrieval[42][43][44][45]. Unsupervised contrastive hashing (UCCH) [30] introduces contrastive learning into hash retrieval and narrows the gap between contrastive learning and hashing with a new momentum optimizer. Transformer-based hamming hashing for efficient image retrieval (TransHash) [46] is the first method to address the deep hash learning problem without the support of convolutional neural networks (CNNs), using a dual-stream framework based on the vision transformer (ViT) for extracting features from images and texts, and utilizes dynamic similarity matrices to learn compact binary codes. With the introduction of the CLIP model, more methods [47][48][49] have begun using this technology for extracting features from images and texts.

Table 3: Partition of Datasets

| Dataset | MIRFlickr25K | NUS-WIDE | MS-COCO |
|---------|--------------|----------|---------|
| Size    | 25,000       | 269,648  | 123,289 |
| Label   | 24           | 81       | 80      |
| Query   | 2000         | 2100     | 5000    |
| Train   | 10000        | 10500    | 10000   |
| Text    | 1386D        | 1000D    | 2026D   |

## B  Datasets

As shown in Table 3, we can see the statistics of each dataset used in our paper.

MIRFlickr25K is a dataset that contains 25,000 images, each with multiple text tags. In addition, each image is annotated with at least one of 24 categories. In our experiments, we selected only those image-text pairs that have at least 20 tags as our experimental data.

NUS-WIDE consists of 269,648 web images with text tags, each annotated with at least one of 81 categories. We selected 186,577 image-text pairs belonging to the ten most common categories.

MS-COCO includes 123,289 images, each annotated with at least one of 80 categories. In our experiments, samples without instances in the text samples were removed.

## C   Evaluation Metrics

Mean average precision (mAP): A standard criterion to evaluate retrieval accuracy. For each query, the average precision (AP) is first calculated, which is the average of precision values at different recall levels. Then, by averaging the AP values across all queries, the mAP value is obtained, reflecting the specific performance of the retrieval.

Precision-recall curve (PR curve): A method that measures the performance of a model at different threshold settings, providing a comprehensive view of the model's performance across various thresholds, and precisely depicting the relationship between precision and recall.

Top-K precision curve (top-K curve): This metric aims to evaluate the accuracy of retrieval in the top N results, to measure the effect of retrieval among different numbers of samples.

