# OpenReview forum: "An End-To-End Graph Attention Network Hashing for Cross-Modal Retrieval"
_NeurIPS.cc/2024/Conference — NeurIPS 2024 poster_

### Official Review · Reviewer_L29L · 2024-07-08

**Soundness:** 3
**Presentation:** 4
**Contribution:** 4
**Rating:** 7
**Confidence:** 5

**Summary:**

In this paper, a cross-modal hash retrieval method based on graph attention networks is proposed. The main framework of the authors' research contains three key components: feature extraction, graph attention classifier and hash code module. An end-to-end architecture is realized by combining CLIP and Transformer techniques, and in this work, while using GAT as a label classifier, the model is able to better capture and utilize the semantic information of the labels and dig deeper into the graph structure features of the data and their interconnections. A hash optimization strategy and loss function are used in the hash code module to further optimize the hash code. It is shown through extensive experiments that the authors' proposed method achieves better performance in the field of cross-modal hash retrieval.

**Strengths:**

1. This thesis provides a new perspective on cross-modal hash retrieval by using more novel techniques combined with each other in the framework composition.
2. The use of GAT as a tag classifier is designed to facilitate better capture of semantic information of tags while achieving more desirable experimental results.
3. The components of the framework in this paper are shown through extensive experiments, proving that each component is indispensable.

**Weaknesses:**

1. The paper could delve deeper into the details of how the CLIP and Transformer models are integrated.
2. Further explanation is needed to elucidate the advantages of the Graph Attention Network compared to other methods.
3. Some typos exist in mathematical formulas. For example: ① In Line 202, $U$ and $V$ should be $\mathbf{U}$ and $\mathbf{V}$. ② "$.$" should be "$\cdot$" in Eq. (14). ③ Many mathematical formulas lack necessary punctuations.

**Questions:**

1. I suggest that the authors elaborate on the results of the experiments in the section of the results of the comparison of the data obtained in detail
2. Could the graphs in Figures 2 and 3 in section 3.4 of the paper be improved?
3. The way of constructing matrices for labels mentioned in the paper increases the computational complexity, so in the method of this paper, constructing the adjacency matrix of labels, which involves steps such as thresholding, will this be more complex than the traditional method? And what is the difference of this approach?

**Limitations:**

The authors have discussed the limitations of the proposed method.

---

> ### Author Rebuttal · Authors · 2024-08-07
>
> Weaknesses:
>
> Q1.The paper could delve deeper into the details of how the CLIP and Transformer models are integrated.
>
> R1: Thank you for your comments. We would added more enough detail on the process of how the CLIP and Transformer models are integrated as follows. In the feature extraction module, an image feature learning network that combines CLIP and Transformer with each other and a text feature learning network based on the attention mechanism of Transformer are used to obtain the feature representations of images and texts.
> R1: Thank you for your comments. More detail are added as follows.
>
> Q2.Further explanation is needed to elucidate the advantages of the Graph Attention Network compared to other methods.
>
> R2: Using GAT to predict labels have many advantages. Compared with labels during training that is static, predicted labels with GAT is dynamic, which can be updated according to specific tasks and data sets to model the relationship between labels explicitly and improve the representation of label features. In new samples, where there is no clear label, the label definition is not clear, or a sample contains multiple labels, the graph structure of GAT can help to infer the label of the sample, so as to obtain better inference and classification. Using a graph structure, GAT reduces the impact of label noise on training data by smoothing operations, while being able to predict labels and reduce label mismatches in original sample.
>
> Q3.Some typos exist in mathematical formulas. For example: ① In Line 202, “U and V” should be “U and V” . ② "." should be "." in Eq. (14). ③ Many mathematical formulas lack necessary punctuations.
>
> R3: Sorry for this, full text would be carefully checked to ensure that the typos
> in mathematical formulas. “U and V” would be changed with “U and V”, "." would be changed with ".".Punctuations of mathematical formulas would be added in our final manuscript.
>
> Questions:
>
> Q1. I suggest that the authors elaborate on the results of the experiments in the section of the results of the comparison of the data obtained in detail
>
> R1: Thank you for your comments. More detail are added as follows.
>      In order to further investigate the performance of our EGATH method, we used three evaluation metrics: mAP, PR curve and Top-k curve, as can be seen from the tables in the paper, we compared with the state-of-the-art seven methods on three datasets, the underlined values in the table are the best results among the compared methods, Table 1 illustrates the results of the mAP for the different lengths of the hash codes (16-bit, 32-bit, 64-bit). bit) of mAP, we can get the following observations
> Firstly, our method achieves better experimental results on the three datasets, with 3.4%, 3.8% and 12.4% improvement on MIR Flickr25K, NUSWIDE, and MSCOCO, respectively, compared to the best results DSPH and DCHMT among the compared methods, especially the effect on MSCOCO dataset has a significant enhancement, which is due to the following reasons The MSCOCO dataset with rich label information provides more contextual information for GAT, which effectively improves the accuracy of label prediction and the consistency of cross-modal features, and indicates that our method has a special advantage in dealing with data-rich datasets with diverse labels.
>     Secondly, the corresponding PR curves can be seen in Fig. 2, from which we can observe that EGATH performs better and obtains higher precision scores at all recall levels. EGATH improves semantic differentiation and consistency across modalities through feature networks, and improves potential semantic associations through tag classification based on graph-attention networks, thus facilitating the cross-media retrieval performance.
>     Finally, in Fig. 3 we compare EGATH with other methods under different Top-k search results, including 50, 100, 200, 500, 1000, 1500, 2000, 2500, 3000, 3500, 4000, 4500, and 5000, and these experiments cover retrieval tasks ranging from small to large scale in order to evaluate the different methods at various retrieval scales. The experimental results show that the EGATH algorithm outperforms other existing algorithms for the cross-media retrieval tasks I→T and T→I under any top-K ordering, especially with higher MAP values for larger K values. This advantage is attributed to the fact that the EGATH method better preserves semantic information during the generation of hash codes and thus performs better in large-scale retrieval.
>
> Q2. Could the graphs in Figures 2 and 3 in section 3.4 of the paper be improved?
>
> R2: Sorry for this. We would improve Figures 2 and 3 in our final manuscript. The version that has been changed is in the PDF attachment.
>
> Q3.The way of constructing matrices for labels mentioned in the paper increases the computational complexity, so in the method of this paper, constructing the adjacency matrix of labels, which involves steps such as thresholding, will this be more complex than the traditional method? And what is the difference of this approach?
>
> R3: Constructing matrices for labels does increase computational complexity than traditional method, which is relatively small. However using GAT to predict labels have many advantages. Compared with labels during training that is static, predicted labels with GAT is dynamic, which can be updated according to specific tasks and data sets to model the relationship between labels explicitly and improve the representation of label features. In new samples, where there is no clear label, the label definition is not clear, or a sample contains multiple labels, the graph structure of GAT can help to infer the label of the sample, so as to obtain better inference and classification. Using a graph structure, GAT reduces the impact of label noise on training data by smoothing operations, while being able to predict labels and reduce label mismatches in original sample.

---

> > ### Comment · Reviewer_L29L · 2024-08-12
> >
> > Thank you for the author's response, which has effectively addressed my concerns. I will keep my score on the paper.

---

### Official Review · Reviewer_V2kH · 2024-07-08

**Soundness:** 3
**Presentation:** 2
**Contribution:** 3
**Rating:** 6
**Confidence:** 3

**Summary:**

The paper mainly tries to address the issues about uncomprehensive feature representation and semantic associations of the existing cross modal retrieval works. In the paper, EGATH, an end-to-end graph attention network hash is proposed. EGATH adopts CLIP to improve the generalization ability in semantic consistency across different data modalities. The experiments are performed on three well-known benchmarks, which show the superior performance of the method on cross-modal retrieval tasks.

**Strengths:**

1. The paper provides a novel end-to-end Graph Attention Hash Network for cross-modal retrieval. By combining CLIP and Transformer, the method can capture global features and ensure semantic consistency of multi-modal data.

2. The contributions of the work are clearly stated. The organization is good. The experimental results are convincing. Some ablation studies are also conducted. The network architecture looks beautiful and clear.

**Weaknesses:**

1. Some grammar, spelling, and symbol errors need carefully check.

2. For the title, ‘Network Hash’ seems uncommon. Please check it.

3. It is suggested to check the use of mathematical symbols. For example, it is difficult to distinguish vector, matrix, elements, etc in the paper.

4. In (10), where is $i$?

5. It is suggested to add an Algorithm table to summarize the training/optimization process.

6. Figure 4 is too small to look.

**Questions:**

See weakness

---

> ### Author Rebuttal · Authors · 2024-08-07
>
> Weaknesses & Questions:
>
> Q1.Some grammar, spelling, and symbol errors need carefully check.
>
> R1: Sorry for this, we will carefully check to ensure that there are no grammatical, spelling, and symbol errors in the final version.
>
> Q2. For the title, ‘Network Hash’ seems uncommon. Please check it
>
> R2: Sorry for this,‘Network Hash’would be revised with ‘Hashing Network’ in the final version.
>
> Q3. It is suggested to check the use of mathematical symbols. For example, it is difficult to distinguish vector, matrix, elements, etc in the paper.
>
> R3: Sorry for this, we will carefully check carefully checked to ensure that the difference between vector, matrix and elements in the final version.
>
> Q4. In (10), where is $i$?
>
> R4: Sorry for this. $i$ is the foot mark of $\hat{S}$ in Eq.(9) and also  $\hat{S}$ in Eq.(10).Correct form would be revised in our final manuscript.
>
> Q5. It is suggested to add an Algorithm table to summarize the training/optimization process.
>
> R5: Thank you for this insightful comment. Algorithm table has been added in PDF attachment with Table 2.
>
> Q6. Figure 4 is too small to look.
>
> R6：Sorry for this, Figure 4 would be revised in our final version. The version that has been changed is in the PDF attachment.

---

> > ### Comment · Reviewer_V2kH · 2024-08-13
> >
> > Thanks for your efforts in the response and my questions are addressed.

---

### Official Review · Reviewer_FNxK · 2024-07-11

**Soundness:** 2
**Presentation:** 2
**Contribution:** 2
**Rating:** 3
**Confidence:** 5

**Summary:**

In this paper, the authors present an E2E graph attention network based hashing method for cross-modal retrieval task. The proposed method adopts CLIP and Transformer to extract the features of images or texts. In addition, it also uses a classifier based on graph attention network to predict labels to enhance cross-modal feature representation. The authors demonstrate the effectiveness of the proposed method by experiments.

**Strengths:**

1.	The idea seems technically sound.
2.	There are some contributions to this field, example the combination of CLIP, Transformer and Graph Attention Network.

**Weaknesses:**

1.	The contribution is not significant. Actually, some hashing methods have explored the use of CLIP, Transformer or graph attention network.
2.	The results are not convincing. Only one method proposed in 2023 is used as a basline for comparison. To my knowledge, many methods of this field are proposed in each year. The authors should compare the proposed method with more recent works.

**Questions:**

1.	The ones in the weaknesses.
2.	The authors use GAT to predict labels. Actually, the samples have labels during training. Is it necessary for the model to predict the labels? In addition, from the framework, I cannot understand how the model makes use of the predicted labels.

**Limitations:**

The authors have addressed the limitations.

---

> ### Author Rebuttal · Authors · 2024-08-07
>
> Weaknesses:
>
> Q1. The contribution is not significant. Actually, some hashing methods have explored the use of CLIP, Transformer or graph attention network.
>
> R1: Thank you for your comments. The main contributions of our work are as follows.
> By combining CLIP and Transformer technologies, an end-to-end architecture is implemented that significantly improves the model’s ability to capture global features of multi modal data, thus ensuring semantic consistency of images and text. Different from other existing works, where CLIP and Transformer are both used for text and image to extract features. While we utilize CLIP coupled with transformer to extract features for image data, and feature extraction via a transformer for text, which can realize lightweight network.
> Predicted labels is used to combined with the feature extracted from the feature modules of image net and text net to enhance feature representation, which also reduce the noise in the original labels during training. We utilize GAT as a label classifier to explore the hidden information in the label to predict labels, which can directly model the label graph to dig the correlation between labels and has higher flexibility than other label classification using preset weights for feature processing.
> Our EGATH was evaluated on three widely recognized benchmark datasets: NUS-WIDE, MIRFlickr25K, and MS-COCO, proving that it has obvious advantages in performance. Experimental results on these datasets show that our EGATH is superior to the current advanced cross-modal hash methods.
>
> Q2. The results are not convincing. Only one method proposed in 2023 is used as a baseline for comparison. To my knowledge, many methods of this field are proposed in each year. The authors should compare the proposed method with more recent works.
>
> R2: Thank you for your comments. We have added four new baselines, which includes the GASKN in 2024 ICDE, CDTH in 2024 IEEE TSVT, SCAHN in 2024 IEEE TKD, and CMGCAH in 2023 IEEE TITS. The experiment result shows that our proposed method has much better performance than the four baselines. More detailed are as follows.
> # Table 1 Additional experimental data
>
> | Task | Method      | MIR FLICKr-25K |     |     | NUSWIDE |     |     | MS COCO |     |     |
> |-------|-------------|-------------------|----|--- |------------|----|-----|-----------|----|----|
> |      |             | 16bit          | 32bit | 64bit | 16bit   | 32bit | 64bit | 16bit   | 32bit | 64bit |
> |      | CDTH(2024)   | 0.7317| 0.7461|	0.7477| 0.6596|	0.6613| 0.6700|	 0.5853| 0.6411| 0.6573|
> |      | SCAHN(2024) |0.8123 | 0.8131 | 0.8336 |	0.6588 | 0.6621 | 0.6669 |	0.6727 | 0.7108 | 0.7528 |
> | I→T  | CMGCAH(2023)| 0.7901| 0.8030|  0.8150| 0.6213| 0.6440| 0.6462  | - | -  | - |
> |      | GASKN(2024) | 0.7610| 	0.7720| 	0.7830| **0.7200**| 	0.7260|0.7470|	0.7210| 0.7360| 0.7410|
> |      | **EGATH(Ours)**  | **0.8411**| **0.8562**| **0.8637**| 0.7191| **0.7345**| **0.7480**| **0.7259**| **0.7688**| **0.7945**|
> |      | CDTH(2024)   |  0.7315|	0.7464 | 0.7503 | 0.6788| 0.6815 | 0.6910 | 0.5846 | 0.6427| 0.6573|
> |      | SCAHN(2024) |  0.7890 |	0.7971| 0.8186|	0.6718| 0.6803| 0.6980| 0.7183|	 0.7504| **0.8093**|
> | T→I  | CMGCAH(2023)| 0.7823| 0.7932|	0.8045| 0.6782|	 0.6801| 0.6844| -| -|	-|
> |      | GASKN(2024) |  0.7450 |	0.7560| 0.7720|	0.7080| 0.7170| 0.7450| 0.7090| 0.7200| 0.7280|
> |      | **EGATH(Ours)**  | **0.8064** | **0.8185**  | **0.8293**  | **0.7270**  | **0.7437**| **0.7539** | **0.7247**  | **0.7729** | 0.8015 |
>
> REFERENCE
>
> [1]	Fengling Li, Bowen Wang, Lei Zhu, Jingjing Li, Zheng Zhang, and Xiaojun Chang. Cross-domain transfer hashing for efficient cross-modal retrieval. IEEE Transactions on Circuits and Systems for VideoTechnology, 2024.
>
> [2]	Meiyu Liang, Yawen Li, Yang Yu, Xiaowen Cao, Zhe Xue, Ang Li, and Kangkang Lu. Structures awarefine-grained contrastive adversarial hashing for cross-media retrieval. IEEE Transactions on Knowledgeand Data Engineering, 2024.
>
> [3]	Weihua Ou, Jiaxin Deng, Lei Zhang, Jianping Gou, and Quan Zhou. Cross-modal generation and paircorrelation alignment hashing. IEEE Transactions on Intelligent Transportation Systems, 24(3):3018–3026,2023.
>
> [4]	Yang Yu, Meiyu Liang, Mengran Yin, Kangkang Lu, Junping Du, and Zhe Xue. Unsupervised multimodal graph contrastive semantic anchor space dynamic knowledge distillation network for cross-media hash retrieval. In 2024 IEEE 40th International Conference on Data Engineering (ICDE), pages 4699–4708.IEEE, 2024.
>
> Question:
>
> Q1. The authors use GAT to predict labels. Actually, the samples have labels during training. Is it necessary for the model to predict the labels? In addition, from the framework, I cannot understand how the model makes use of the predicted labels.
>
> R1: Thank you for your comments. Compared with labels during training, the predicted labels can explore the information hidden in labels during training to enhance feature representation. Moreover, labels during training have a lot of noise and mismatches, while predicted labels can effectively handle noise to avoid the mismatches. The model integrates the co-occurrence information from the labels during training into the word embeddings to form the predicted labels (text and images). Then, model combined the predicted labels with the feature extracted from the feature modules of image net and text net. The whole process can not only bridge the information gap between text and image but also enhance the deep characteristics of them.

---

> > ### Comment · Reviewer_FNxK · 2024-08-13
> >
> > Thanks for the authors' responses. The authors have added new results. The authors claim that labels have a lot of noise and mismatches and the predicted labels can effectively hand noise to avoid the mismatches. I cannot agree with this. Usually, we assume that the labels are correct. Moreover, there is no result to support this.

---

> > > ### Author Response · Authors · 2024-08-14
> > >
> > > Although you believe that labels should be assumed to be accurate, this assumption usually only applies when the dataset labels are of high quality and have been scrutinised. However, in real training, label noise and missing labels are often unavoidable, and further processing of the labels is required to capture the hidden semantic information inside the labels.
> > >
> > > Literature [1] details how to perform semi-supervised learning via GCNs and use a small amount of label and graph structure information to improve classification performance, which provides theoretical support for the use of GNNs in the presence of label noise. In the literature [2], Qian et al. proposed ALGCN, which uses a GCN to act as a label classifier in label representation learning, extracting information from the labelled graph to guide the model in learning discriminative features. Literature [3][4][6] demonstrated studies applying GCN in different modalities for label classification prediction, highlighting the ability of GCN in processing labels and exploring the implicit semantic information of labels to enhance the feature representation rather than just assuming that the labels are accurate. Finally, in the literature [5], Duan et al. used GAT for cross-modal tag classification prediction to extract more fine-grained features from tags, thus improving retrieval performance.
> > > These results show that our method has a solid theoretical foundation, and that the proposed method is able to exploit the relationships and common features between different modalities for tag classification, optimise the fusion and matching of cross-modal information, thus improving retrieval accuracy, and is feasible in dealing with real-life tag noise problems.
> > >
> > >
> > > [1]Thomas N Kipf and Max Welling. Semi-supervised classification with graph convolutional networks.arXiv preprint arXiv:1609.02907, 2016
> > >
> > > [2] Shengsheng Qian, Dizhan Xue, Quan Fang, and Changsheng Xu. Adaptive label-aware graph convolutional networks for cross-modal retrieval. IEEE Transactions on Multimedia, 24:3520–3532, 2021.
> > >
> > > [3] Zhao-Min Chen, Xiu-Shen Wei, Peng Wang, and Yanwen Guo. Multi-label image recognition with graphconvolutional networks. In Proceedings of the IEEE/CVF conference on computer vision and patternrecognition, pages 5177–5186, 2019
> > >
> > > [4]Cong Bai, Chao Zeng, Qing Ma, and Jinglin Zhang. Graph convolutional network discrete hashing forcross-modal retrieval. IEEE Transactions on Neural Networks and Learning Systems, 2022.
> > >
> > > [5] Youxiang Duan, Ning Chen, Peiying Zhang, Neeraj Kumar, Lunjie Chang, and Wu Wen. Ms2gah: Multi-label semantic supervised graph attention hashing for robust cross-modal retrieval. Pattern Recognition,128:108676, 2022.
> > >
> > > [6]Meiyu Liang, Yawen Li, Yang Yu, Xiaowen Cao, Zhe Xue, Ang Li, and Kangkang Lu. Structures awarefine-grained contrastive adversarial hashing for cross-media retrieval. IEEE Transactions on Knowledge and Data Engineering, 2024

---

### Official Review · Reviewer_5SGA · 2024-07-12

**Soundness:** 3
**Presentation:** 4
**Contribution:** 3
**Rating:** 6
**Confidence:** 5

**Summary:**

This paper proposes a method for cross-modal hash retrieval, unlike previous methods, the EGATH proposed in this paper combines multiple techniques with each other, which can effectively improve the performance of retrieval. At the same time, graph attention network is introduced to further process the information inherent in labels, which can enable better mining of potential semantic associations between different modalities. In addition, EGATH also introduces hash code optimization strategies to further preserve the semantic relationships of hash codes.
In conclusion, the method proposed by the authors and utilizing graph attention networks as a label classifier seems to be feasible.

**Strengths:**

1.	This paper is clearly written and easy to understand.
2.	This paper proposes a novel approach for the application of graph attention networks in cross-modal retrieval.
3.	The paper has strong empirical evidence to support the effectiveness of the proposed method.

**Weaknesses:**

1. the paper does not provide enough detail on some specific components of the framework.
2. Some of the baselines could be added to this paper as appropriate.

**Questions:**

1. The authors need add some details about the graph attention network module, how to process the components.
2. In this paper, does the use of CLIP and Transformer and GAT models involve high model complexity? Does this affect the training performance of the model?
3. The author need add comparative experiments on Graph Attention Networks to further illustrate the importance of this module.

**Limitations:**

This paper emphasizes the limitations of the proposed methodology.

---

> ### Author Rebuttal · Authors · 2024-08-07
>
> Weaknesses:
>
> Q1. the paper does not provide enough detail on some specific components of the framework
>
> R1:We would add more enough detail on the framework. we propose a new end-to-end cross-modal hash retrieval based on graph attention network, which contains three main components: feature extraction, GAT-based classification module and hash learning module. Firstly, in the feature extraction module, an image feature learning network that combines CLIP and Transformer with each other and a text feature learning network based on the attention mechanism of Transformer are used to obtain the feature representations of images and texts, and secondly, in the GAT-based tag prediction classifier, we integrate the co-occurrence information of tags into word embeddings, specifically, calculate the co-occurrence frequency of the labels and construct the adjacent matrix, take the initial feature vector and adjacent matrix of the labels as inputs, mine the implicit semantic information present in the labels through the dynamic attention mechanism, and compute the attention weights between the labels, so as to weight the average and update the embedding of the labels, and the updated embedding of the labels not only contains the information of the labels themselves, but also integrates the relational information between the labels, which is used to guide the feature learning of images and texts. Finally, the hash code loss function and optimum strategy enable the hash code to retain more semantic information, thus obtaining a discriminative and compact cross-modal uniform hash representation.
>
> Q2. Some of the baselines could be added to this paper as appropriate
>
> R2:  We have added four new baselines. The experiment result shows that our proposed method has much better performance than the four baselines.
> | Task | Method      | MIR FLICKr-25K |     |     | NUSWIDE |     |     | MS COCO |     |     |
> |-------|-------------|-------------------|----|--- |------------|----|-----|-----------|----|----|
> |      |             | 16bit          | 32bit | 64bit | 16bit   | 32bit | 64bit | 16bit   | 32bit | 64bit |
> |      | CDTH(2024)   | 0.7317| 0.7461|	0.7477| 0.6596|	0.6613| 0.6700|	 0.5853| 0.6411| 0.6573|
> |      | SCAHN(2024) |0.8123 | 0.8131 | 0.8336 |	0.6588 | 0.6621 | 0.6669 |	0.6727 | 0.7108 | 0.7528 |
> | I→T  | CMGCAH(2023)| 0.7901| 0.8030|  0.8150| 0.6213| 0.6440| 0.6462  | - | -  | - |
> |      | GASKN(2024) | 0.7610| 	0.7720| 	0.7830| **0.7200**| 	0.7260|0.7470|	0.7210| 0.7360| 0.7410|
> |      | **EGATH(Ours)**  | **0.8411**| **0.8562**| **0.8637**| 0.7191| **0.7345**| **0.7480**| **0.7259**| **0.7688**| **0.7945**|
> |      | CDTH(2024)   |  0.7315|	0.7464 | 0.7503 | 0.6788| 0.6815 | 0.6910 | 0.5846 | 0.6427| 0.6573|
> |      | SCAHN(2024) |  0.7890 |	0.7971| 0.8186|	0.6718| 0.6803| 0.6980| 0.7183|	 0.7504| **0.8093**|
> | T→I  | CMGCAH(2023)| 0.7823| 0.7932|	0.8045| 0.6782|	 0.6801| 0.6844| -| -|	-|
> |      | GASKN(2024) |  0.7450 |	0.7560| 0.7720|	0.7080| 0.7170| 0.7450| 0.7090| 0.7200| 0.7280|
> |      | **EGATH(Ours)**  | **0.8064** | **0.8185**  | **0.8293**  | **0.7270**  | **0.7437**| **0.7539** | **0.7247**  | **0.7729** | 0.8015 |
>
> [1] Cross-domain transfer hashing for efficient cross-modal retrieval. IEEE Transactions on Circuits and Systems for Video Technology, 2024
>
> [2] Structures awarefine-grained contrastive adversarial hashing for cross-media retrieval. IEEE Transactions on Knowledge and Data Engineering, 2024
>
> [3]Cross-modal generation and pair correlation alignment hashing. IEEE Transactions on Intelligent Transportation Systems, 24(3):3018-3026,2023
>
> [4]Unsupervised multimodal graph contrastive semantic anchor space dynamic knowledge distillation network for cross-media hash retrieval. In 2024 IEEE 40th International Conference on Data Engineering (ICDE), pages 4699-4708
>
>  Questions:
>
> Q1. The authors need add some details about the graph attention network module, how to process the components.
>
> R1: We would added the following detail in our final manuscript.
>
> Firstly, the GloVe vectors and adjacency matrices of the labels are taken as inputs, and the information of the neighbour nodes is aggregated through the attention mechanism of GAT, which maps the input features to a new feature space by linearly transforming the input features using the weighting matrix. Secondly, each pair of labels is spliced and the attention score is computed to enable the model to assign higher weights to important node information, which in turn updates the feature representation of each label by weighted aggregation of information from neighbour nodes using the attention score. Finally, the updated label features are fused with the input features (image or text features), and the similarity is calculated by matrix multiplication to obtain the final category prediction.
>
> Q2. In this paper, does the use of CLIP and Transformer and GAT models involve high model complexity? Does this affect the training performance of the model.
>
> R2:  Constructing matrices for labels does increase computational complexity than traditional method, while this doesn’t affect the training performance of the model. Actually, using GAT as a label classifier to predict labels can provide more context information. Moreover, combining the predicted labels with features of the extracted image and text can enhance the representation ability of feature representation, at the same time, improve the consistence and matching effect between images and text.
>
> Q3.The author need add comparative experiments on Graph Attention Networks to further illustrate the importance of this module.
>
> R3:We conducted further tests for this module in the added comparison experiments, specifically using the GASKN model from the 2024 ICDE conference. The experimental results show that the addition of the graph network module significantly improves retrieval accuracy, which validates the effectiveness of graph networks in improving cross-modal retrieval performance.

---

> > ### Comment · Reviewer_5SGA · 2024-08-12
> >
> > Thanks for the author's response. I have no further questions and will maintain my previous decision.

---

### Author Rebuttal · Authors · 2024-08-07

Dear Reviewers,

We appreciate the time and effort you have put into reviewing our paper titled “[An End-To-End Graph Attention Network Hash for Cross-modal Retrieval].” We value your feedback and have addressed each of the comments raised. Below, we provide detailed responses to the reviewers' comments.   For the relevant figures, tables, and algorithms, please view the attached PDF file.

---

### Comment · Area_Chair_Pso4 · 2024-08-14

Dear all,

Thanks for your time and efforts in reviewing this paper. This is the right and emerging time to discuss this paper with the authors.

The authors provided their rebuttal, and some reviewers have posted their partial discussion about this paper.

Any discussion is welcome and you may consider reading each others' reviews, posting a question, and reaching a consensus.

Best,
Your AC

---

### Decision · Program_Chairs · 2024-09-25

**Decision:**

Accept (poster)

**Comment:**

This paper adopts the CLIP combined with the Transformer to improve understanding and generalization ability in semantic consistency across different data modalities. Several experiments on benchmark datasets show that they significantly perform favorably against several methods. Three of four reviewers rate the paper as weak accept/accept, while one reviewer still has concerns about the contributions and state-of-the-art performance. After discussion, the final decision is accept, and the authors should further highlight innovations and improve experiments.